# Effect of temperature on sporulation and spore development of giant kelp (*Macrocystis pyrifera*)

Duong M. Le [1,2][�]*, Mathew J. Desmond[1,2][�], Daniel W. Pritchard[2], Christopher D. Hepburn[1,2]

**1** Department of Marine Science, University of Otago, Dunedin, New Zealand, **2** Coastal People Southern Skies Centre of Research Excellence, University of Otago, Dunedin, New Zealand

euro These authors contributed equally to this work.

* ledu229p@otago.ac.nz

**Data Availability Statement:** The data is now available on the figshare database under https://doi.org/10.6084/m9.figshare.20960476.v1.

## Abstract

Rising ocean temperature is a major driver of kelp forest decline worldwide and one that threatens to intensify over the coming decades. What is not particularly well understood are the mechanisms that drive loss and how they operate at differing life stages. This study aimed to establish an understanding of the effects of increasing temperature on the early developmental stages of the giant kelp, *Macrocystis pyrifera*. Sporulation was carried out across 10 temperature treatments from 9.5 to 26.2°C ± 0.2°C at approximately 2°C intervals. Spores were then incubated at these temperatures under a 20.3±1.7 μmol photons m$^{-2}$ s$^{-1}$, 16L:8D photoperiod for 5 days. Results indicate that spore release was positively correlated with increasing temperature, whereas an inverse trend was observed between temperature and the growth of germ-tube. The thermal threshold for spore and germling development was determined to be between 21.7°C and 23.8°C. Spore settlement was the most drastically effected developmental phase by increasing temperature. This study highlights the vulnerability of early life stages of *M. pyrifera* development to rising ocean temperature and has implications for modelling future distribution of this valuable ecosystem engineer in a changing ocean.

## Introduction

Kelp forests represent one of the most diverse and productive natural ecosystems on the planet [1]. Their temperate bi-hemispheric distribution means they support many of the world's large coastal fisheries through the provision of food and habitat [2–4] and through their physical presence they offer many ecosystem services [5, 6]. They are comprised predominantly of fleshy brown algae, with the dominant species being from the order Laminariales [3]. They are commonly inhabited by a wide diversity of other algal species of Phaeophyceae, Chlorophyceae and Rhodophyceae. For centuries, humans have relied on these systems for the range of social and economic benefits they offer [5, 6], however in the past 50–100 years significant swaths of

**Funding:** This study was funded by New Zealand Ministry of Business, Innovation and Employment to MJD and CDH (grant number UOOX1908 - "Cultivating resilient marine forests to rebuild productive coastal ecosystems"). DML was awarded a postgraduate scholarship from University of Otago, New Zealand. Website of New Zealand Ministry of Business, Innovation and Employment: https://www.mbie.govt.nz/ Website of University of Otago: https://www.otago.ac.nz/ The funders had no role in study design, data collection and analysis, decision to publish, or preparation of the manuscript.

**Competing interests:** The authors have declared that no competing interests exist.

kelp forest have been lost and many of the remaining systems show a declining trajectory [1, 7–10].

A wide range of factors are responsible for global kelp forest decline, with most of them being the result of anthropogenic activities [10]. Increasing ocean temperature is a global stressor that is negatively influencing the distribution and productivity of kelp dominated ecosystems at the warm leading edges of their range [11–13]. Temperature plays a pivotal role in the physiology, ecology and the geographical distribution of kelp forest species. However, the mechanistic effects by which temperature exerts this controlling effect is often poorly understood [14, 15]. Elevated temperature is known to influence photosynthetic performance and rates of growth, at elevated levels it can increase tissue loss and deterioration [16, 17]. It also plays a critical role in controlling the presence of other species that may predate directly on kelp such as mesograzers, bryozoan, and sea urchins [17–19]. With a rate of increase of 0.13˚C per decade for the past thirty years and a predicted total increase of 1–3˚C by the turn of the century [20], global sea surface temperature has, and will continue to, negatively influence many of the world's kelp forest ecosystems.

*Macrocystis pyrifera* is arguably one of the most iconic kelps, creating the largest biogenic system of all kelp forest forming species [21–24]. It is also a species that holds significant value for human use, for direct consumption and for its valuable extracts [25, 26]. *M. pyrifera* is no exception to the global trend of decline and has undergone significant loss in many regions as a result of increasing temperature [9, 10, 27–31]. The optimum thermal range for *M. pyrifera* growth is estimated to be from 12–17˚C [32] and it is thought that the maximum thermal threshold for survival ranges from 18–25˚C [27, 33–36]. This range in the maximum thermal threshold likely results from local acclimation, potential genetic adaptation, and the availability of other resources such as light and nutrients [37–40].

Like all kelps, *M. pyrifera* has a bi-phasic life cycle with alternating generations between the microscopic haploid phase and the macroscopic diploid phase [41]. The effects of elevated temperature on the physiology of the macroscopic sporophyte stage are relatively well researched [35, 36, 39, 40, 42–45], but the effect on the earlier microscopic life stages, such as the spore, germling and gametophyte, remain poorly understood [46–48]. These early life stages may hold important information that will provide a greater understanding of how rising temperatures are negatively influencing *M. pyrifera* and insight into how populations will fare under predicted climate change scenarios. This information will also be important for selecting, and potentially thermally priming [49], progeny for use in restoration and aquaculture applications to help encourage greater thermal resilience.

The focus of this study was to quantify the effects of temperature on the early developmental stages of *M. pyrifera*, from sporulation through to germination. *Macrocystis pyrifera* spores were released and their survivorship, germination and growth measured over five days and across ten temperature treatments ranging from 9.5 to 26.2 ± 0.2˚C. The range of temperatures tested span the common thermal range of *M. pyrifera* globally and exceeds the currently recognised thermal threshold estimates. This information is important for clarifying at which life stage *M. pyrifera* is most vulnerable to elevated temperatures and can be used for predicting the persistence of this important species under future climate scenarios.

## Material and methods

### Site selection and tissue collection

Sori from six *M. pyrifera* individuals, 10–20m apart, were collected in October 2020, using SCUBA from Otago Harbour (45˚47'45.9"S 170˚39'20.2"E), New Zealand. The site was sheltered with hard rocky substrate, the depth was 2-3m, and seawater temperature at collection

sites was 12.5˚C (austral spring). Samples were placed in a plastic zip-lock bag with ambient seawater, kept in the dark in a cooler bin and returned to laboratory within four hours.

## Pre-treatment

Upon return to the laboratory, the sorus tissue was gently rinsed with filtered seawater (0.22 μm, Millipore) and visible epiphytes removed using damp paper towels. The sori were then wrapped in moist tissue paper, covered with aluminium foil and left in a refrigerator at 4˚C for 24 hours.

## Heat block system and temperature levels

A large aluminium thermal block system (L x W x H: 77 x 32 x 6 cm) was designed to allow for six replicates at each temperature level, one for each the individual sori collected. To create a temperature gradient, one end of the heat block was connected to a cool water bath and the other end to a hot water bath and water circulated through a piped system allowing conduction across the block. The system allowed for 12 temperature treatments, however only ten were used for this experiment (9.5, 11.1, 12.9, 14.8, 16.3, 18.1, 19.8, 21.7, 23.8, and 26.2˚C ± 0.2˚C). This temperature range was selected to reflect the broad range of sea surface temperatures experienced across the distribution of *M. pyrifera* globally as well as to breach the current known upper thermal threshold for this species [27, 33–36, 50]. To reduce temperature variance the heat block was allowed to stabilise for five days prior to conducting the experiment. Over the duration of the experiment the temperature varied ± 0.2˚C per well. Light was provided via a white fluorescent tube light at 20.3 (±1.7) μmol photons $m^{-2}$ $s^{-1}$.

## Effect of temperature on spore release

In order to prepare for sporulation, 20 mL of Provasoli enriched autoclaved-filtered (0.2 μM, Millipore) seawater (PES) [51] was pipetted into 60 glass vials (bottom surface 615.4 $mm^2$, total volume 30 mL). Vials were placed in the heat block 24 hours prior to sporulation in order to reach the desired treatment temperatures.

After 24 hours in the refrigerator, the sorus tissue was removed and left at room temperature (12˚C) for 30 minutes. The sorus from each individual was then cut into 1 cm x 1 cm pieces and a total of two pieces (2 sided sorus = 4 $cm^2$) were placed in each vial. After 30 min the tissue was removed from each vial, the vial agitated and 1 mL of the spore solution was taken immediately to calculate spore production. The number of spores was assessed using a Neubauer-improved haemocytometer (Marienfeld, Germany) under light microscope (Olympus BX51TRF, Japan) at 20X magnification. For every vial, three aliquots of spore solution were used to count spores, and the mean of these aliquots used as a measure of the number of spores per vial (i.e. every vial was considered as a replicate, n = 6). This was used to calculate spore production (spores per square centimetre of sorus tissue) and the maximum potential settlement (spores per square millimetre of vial basal area). For the duration of experiment light was provided as mentioned above and set to a 16L:8D cycle.

## Effect of temperature on spore development

The experiment was carried out over a five-day period and the temperature in each vial was measured every second day using an infrared thermometer (Fluke, USA). All spores that could settle, were considered to have done so after 24 hours. This was based on settling rates from previous work [52] and applied to the geometry of the vial. After this time the medium in each vial was completely replaced to discard any non-viable spores or those that had not settled and

would interfere with photographic analysis [52]. At days one, three and five, a photograph was taken from three randomly chosen fields of view (FOV) in each vial using a Digital camera MC4K (Microscope X, China) mounted on an inverted microscope (Nikon eclipse Ts2, Japan) at a lens magnification of 20X. Images were analysed using Fiji software version 2.1.0/1.53c [53] and means of the three randomly chosen fields of view were used to calculate spore settlement, germination rate, cell survival and growth rate of gametophytes.

**Settlement.** Spores were considered settled if attached to the bottom of the vial. The percentage of spores settled was calculated by dividing the total number of settled cells (including germinated and non-germinated spores) observed by the maximum potential settlement calculated from spore release on day zero, multiplied by 100.

**Germination.** Germination was calculated at days one, three and five by quantifying the proportion of germinated and non-germinated spores in each vial. Only spores that had a germ-tube greater or equal to 1.5 μm were classified as germinated [54].

**Survivorship.** Cell survivorship was estimated as a percentage by diving the total number of cells (including gametophytes and spores) at day five by the total number observed at day one and multiplying by 100. Viable cells were classified as those which had not lost their shape, even if some depigmentation was observed [35].

**Germ-tube growth.** Growth of germ-tube was measured at day one and day five. In each field of view, 10 randomly selected germlings were measured (overall length, μm) for the 9.5 to 21.7˚C temperature treatments. For the 23.8˚C temperature treatment, low germination rate and survivorship meant only 5 germlings could be measured in each field of view. No cells survived in the highest temperature, so this treatment was removed from this analysis. In each case, the mean of these measurements at each timepoint was used to calculate growth rate (n = 6).

**Statistical analyses.** A linear mixed-effects modelling (LLM) approach was applied using lme4 package [55] in R version 4.0.3 [56]. Preliminary analysis using alternate generalized linear models confirmed that a LLM approach (Gaussian family with an identity link) was appropriate for these data, achieving lowest Akaike Information Criteria. Temperature and where possible (i.e., germination rate and germ-tube growth) time (experimental duration / days 1 to 5) were included as categorical fixed effects, and individual (sori of *M. pyrifera*, n = 6) was included as a categorical random effect. Analysis of deviance was conducted with the default type II error and Wald Chisq test using the *car* package [57]. Multiple comparisons among fixed factors were conducted using Tukey test in the *rcompanion* package [58].

## Results

### Spore production

Spore production increased with rising temperatures (Fig 1A). Mean spore production ranged from $1.97 \pm 1.27 \times 10^5$ at 9.5˚C to $35.97 \pm 17.04 \times 10^5$ spores cm$^{-2}$ sporophyll at 26.2˚C. Spore production at 26.2˚C was significantly greater than at temperatures 21.7˚C and below (p-value < 0.001, Fig 1A).

### Settlement

Spore settlement (mean % ± SD percentage points (p.p.)) 24 hours post sporulation ranged between 3.35% (± 3.82 p.p.) and 46.80% (± 14.32 p.p.) with the lowest settlement occurring at 26.2˚C and the greatest settlement occurring at 21.7˚C (Fig 1B). Spore settlement ranged between approximately 25 to 50% of initial spore densities for treatments from 9.5 and 23.8˚C (Fig 1B).

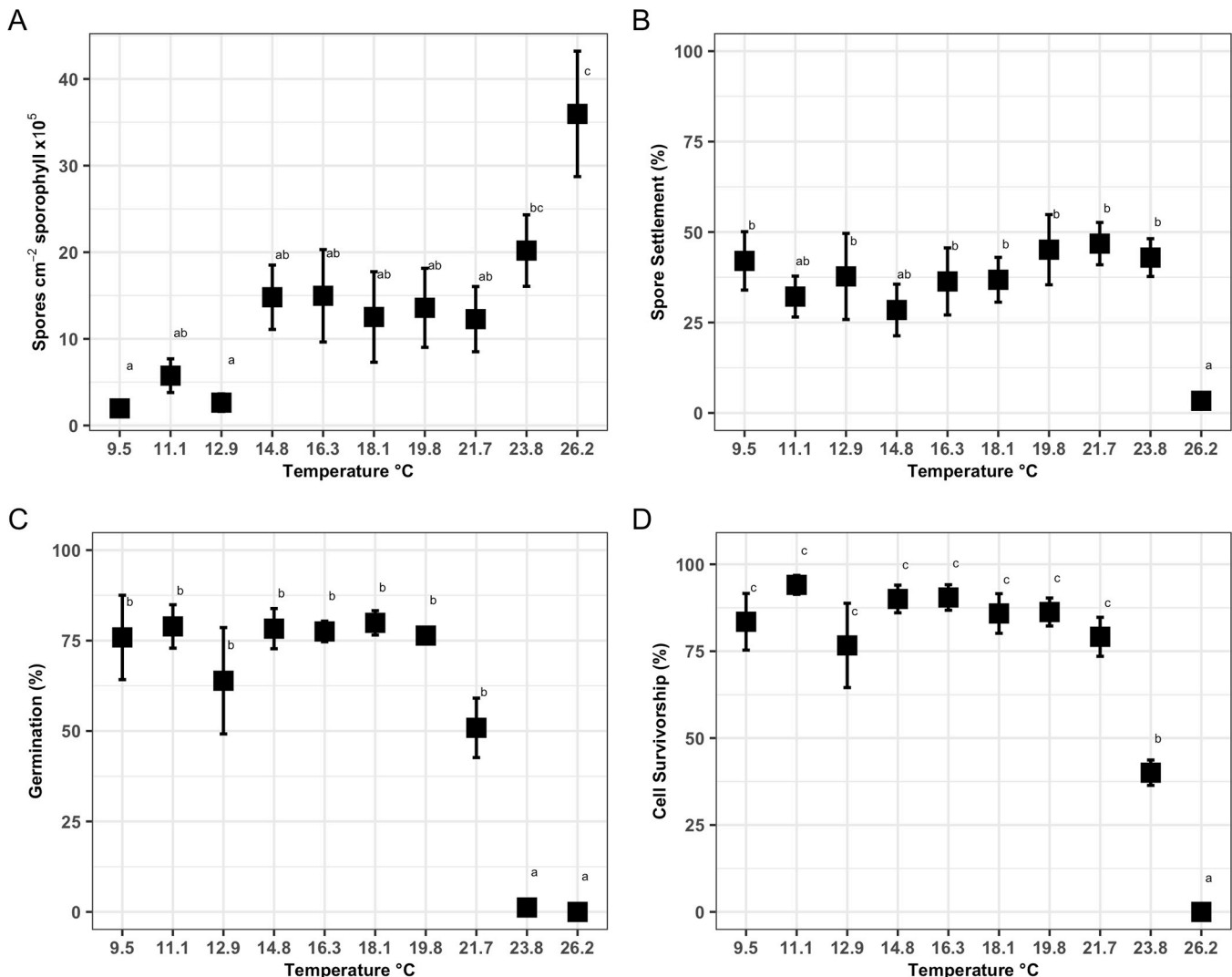

**Fig 1. Physiological responses of the early life stages of *M. pyrifera* across 10 temperature treatments from 9.5–26.2 ± 0.2°C.** A: Mean (± SE) spore production after 30 min incubation time, n = 6; B: Mean (± SE) percentage of spore settlement 24 hours post sporulation, n = 6 (except treatment 12.9°C: n = 5); C: Mean (± SE) percentage of settled spores germinated of *M. pyrifera* after five days, n = 6 (except treatment 12.9°C: n = 5). D: Mean (± SE) percentage cell survivorship after five days, n = 6 (except treatment 12.9, 16.3, 18.1, 21.7°C: n = 5; 19.8°C: n = 3). Letters indicate statistical significance between treatments (Tukey-adjusted, p-value<0.05).

## Germination

Germination success ranged between 0 and 79.89 ± 8.25% with the highest percent germination occurring at 18.1°C and the lowest at 26.2°C (Fig 1C). Germination success was significantly greater at temperatures 21.7°C and below, (p-value < 0.001, Fig 1C). Above 19.8°C spore germination decreased significantly and was ~1% above 23.8°C (p-value < 0.001, Fig 1C) with the exception of 21.7°C. There was no significant interaction between temperature and time on the percent of germination success (p-value = 0.99).

## Cell survivorship

Cell survivorship (the total spore and gametophyte count) after five days ranged between 0 and 94.08 ± 6.67% (Fig 1D). Complete mortality of cells occurred at 26.2°C and nearly 60% at

23.8˚C, with both treatments having significantly lower survivorship than all other temperature treatments (p-value <0.001; Fig 1D). Approximately 98% of viable cells under the 23.8˚C treatment had not germinated and remained at the spore stage.

### Germ-tube growth

The greatest germ-tube diameter on day one were observed between the treatments of 12–19˚C (Fig 2). Average germ-tube diameter at day five was significantly greater than that at day one for all treatments below 21.7˚C (p-value < 0.001, Table 1), with the greatest length being 17.5 ± 1.3 μm and 21.7 ±1.3μm at 14.8˚C from day one and day five respectively. The greatest increase in diameter between day one and day five was observed in the 9.5 and 11.1˚C treatments (Fig 2). There was a significant interaction effect between temperature and time on germ-tube growth (p-value < 0.001, Table 1). No germ-tube growth was recorded in the 26.2˚C treatment as all cells had perished.

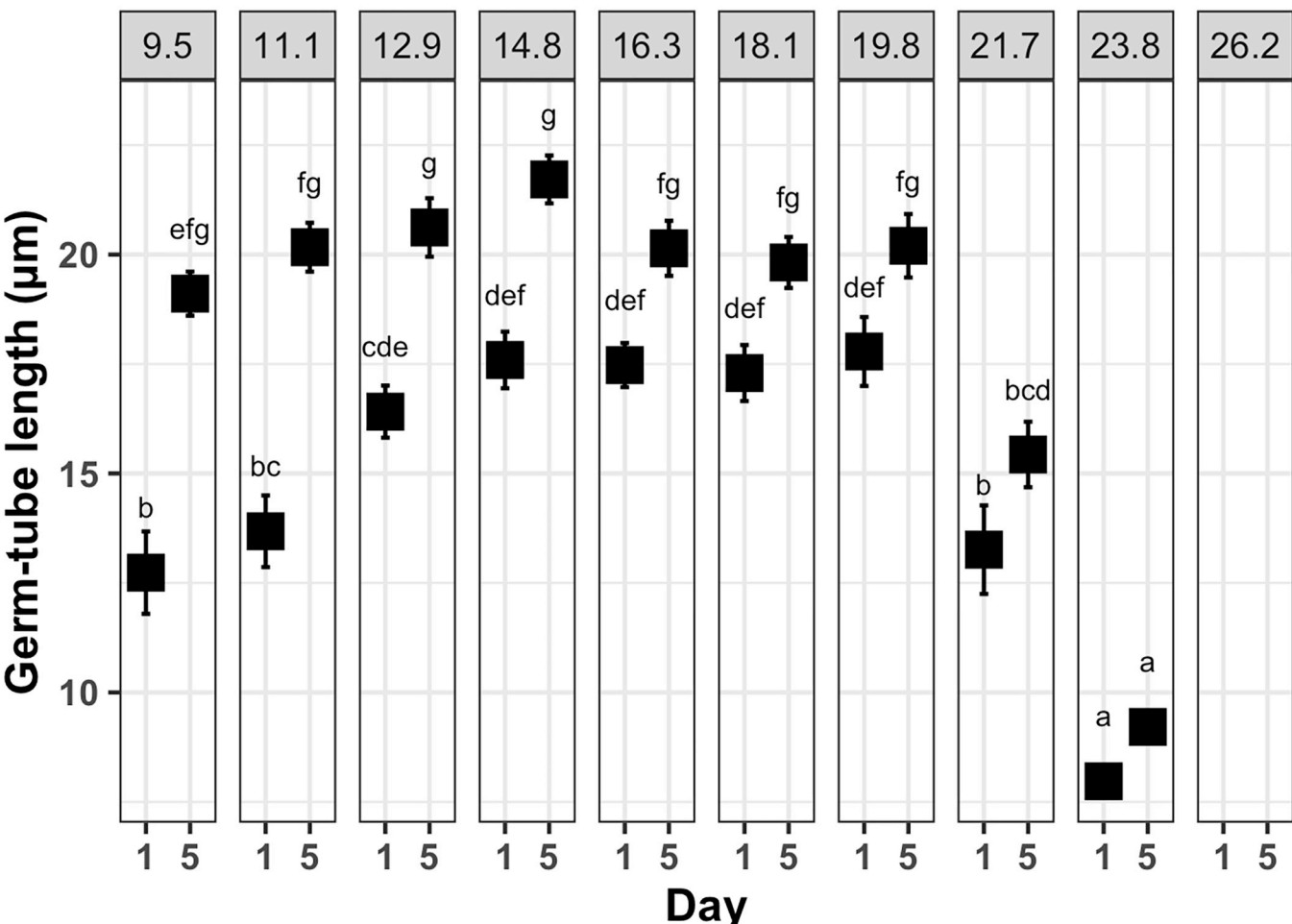

**Fig 2. Mean (± SE) length of germ-tube (μm) of germinated spores of *M. pyrifera* in each of two sampling points with ten different temperature treatments from 9.5–26.2 ± 0.2˚C.** Letters indicate statistical significance between treatments and time (Tukey-adjusted, p-value<0.05), n = 6, (except treatment 12.9˚C: n = 5).

**Table 1. Information of fitted models using a linear mixed-effects modelling (LLM) with a Gaussian family and identity link.** Temperature and time (experimental duration, in day) were fitted as fixed effects, and individual was random effect. Statistical significance at *p-value* < 0.05.

| Responses | Main effect | Df | Chisq | *p-value* |
|---|---|---|---|---|
| Spore production | Temperature | 9 | 63.7 | < 0.001 |
| Spore settlement | Temperature | 9 | 32.1 | < 0.001 |
| Survivorship | Temperature | 9 | 260.4 | < 0.001 |
| Germination | Temperature | 9 | 725.8 | < 0.001 |
| | Time | 2 | 5.4 | 0.07 |
| | Temperature*Time | 18 | 4.2 | 0.99 |
| Germ-tube growth | Temperature | 8 | 594.9 | < 0.001 |
| | Time | 1 | 171.8 | < 0.001 |
| | Temperature*Time | 8 | 42.2 | < 0.001 |

## Discussion

Ocean warming is one of the most significant drivers of global kelp forest decline and is a threat that will continue to increase over the coming decades [28, 30, 59, 60]. This study highlights the thermal tolerance and effect of increasing temperature on the early developmental stages of the globally ubiquitous kelp *M. pyrifera* from a population in southern New Zealand. Although increasing temperature showed a positive relationship with spore production, the effect of exposure to elevated temperatures above 19.8°C was negative for other key developmental parameters. Temperatures above 19.8°C are not uncommon at the warm leading edge of many *M. pyrifera* populations, particularly during heatwave events which are becoming more frequent [61]. Few studies have focused on these early life stages of *M. pyrifera* [46–48] with most being concerned with the more advanced haploid or diploid life stages [35, 36, 39, 42, 43]. This information is essential in order to accurately predict the overall effect of rising ocean temperatures on such an important ecosystem engineer and the higher trophic level species it supports.

Previous work has shown temperature to be an influential factor for spore release. From populations in Chile it was found that spore production was greatest at 15°C when measured across three temperature treatments 8, 15 and 18°C [44], while from the population sampled in this study an increase in spore release was observed up until the maximum treatment temperature of 26.2°C. This finding raises a number of questions regarding how increased temperatures in natural systems may influence the population dynamics. It is unknown whether increased spore production in *M. pyrifera* can compensate the negative effects of rising temperature. Few studies have focussed on the impact of temperature on spore release in macroalgae [62]. At least two potential scenarios exist, the first being that elevated temperatures cause the mass release of spores which are negatively impacted by elevated temperatures, with no net benefit to *M. pyrifera* populations. A second scenario may be that increased temperature transiently increases spore delivery to the substrate under a scenario where temperature then drops back below harmful thresholds, which might be a net benefit to *M. pyrifera* recruitment. The manipulation of temperature may also be of interest in an aquaculture setting where an increase in spore release could be favourable for early seeding of stock. That said, the effect of increased temperature, even for a short period of time, on the later development of *M. pyrifera* is unknown and should be the focus of further research.

Settlement success is key in determining the survival and abundance of kelp propagules, and therefore the structure of adult populations [63–66]. Many factors are known to influence settlement rate and success such as nutrient concentration [67], presence of other biota [42], light condition [68] and sediment cover [69], but until now very little has been known

regarding the effect of temperature. In this study, the percentage of spore settlement was nega-
tively impacted by increasing temperature and significantly declined above 23.8˚C. Interest-
ingly, although 100% of cells perished in the 26.2˚C treatment after day three, ~30% survived
at 23.8˚C but did not germinate. A possible explanation is that the haploid stage of *M. pyrifera*
may show greater thermal tolerance than the adult sporophyte and could potentially remain
dormant under elevated temperatures [32, 35, 43, 70]. In addition, the haploid stage of *M. pyri-
fera* could tolerate, and fully recover (90% after 8 weeks) from, temperatures as high as 24˚C
[35]. What remains unknown is how long these cells can survive under such temperatures,
whether normal germination would occur if temperature was reduced to optimal levels and if
there would be any ongoing impacts to later development from prior exposure. This thermal
window may provide an approach for selecting more thermally tolerant strains of kelp by
exposing cultured progeny to elevated temperatures to remove those that are not thermally
resilient and then reducing the temperature to continue the culturing process. This area of
research warrants further investigation.

Regarding spore germination, the success of germination declined significantly at tempera-
tures above 21.7˚C, below this, no significant effect of temperature was obvious. Interestingly,
1.2 ± 0.5% of spores under the 23.8˚C treatment underwent germination. This finding may
infer that a very small percentage of the spore cohort has naturally higher thermal tolerance
than the rest [35, 36, 39]. This phenomenon likely results from natural genetic variation within
the cohort but highlights the possibility of selecting thermally tolerant progeny for use in resto-
ration or aquaculture applications [9, 36, 71, 72]. The search for thermally tolerant strains of
species is becoming an ever-increasing endeavour as the effects of ocean warming increase
[36]. Further work is needed in this field but for a species such as *M. pyrifera* that has a life
cycle with alternating generations there may be possibilities of identifying resilience at differ-
ent developmental stages.

The length of germ-tube was not significantly affected by temperatures from 9.5 to 19.8˚C
after five days but was significantly reduced above 19.8˚C, indicating that a temperature bottle-
neck was observed between 19.8 and 23.8˚C for germ-tube elongation. It remains unknown
whether the length of germ-tube may influence the fitness of later life stages. A longer experi-
mental duration may address this question.

Marine heatwave events have been recorded in some parts of central and southern New
Zealand where *M. pyrifera* is found exceeding 23˚C in recent years [70]. The coverage of *M.
pyrifera* beds near the site of collection in this study has decreased dramatically in the past five
years due to abnormally high sea surface temperatures in combination with poor water clarity
[31]. Similar heatwaves and declines in *M. pyrifera* populations have been reported elsewhere
including, for example, Tasmania [28], Baja California [30, 43] and British Columbia [73]. It is
predicted that marine heatwaves will occur more frequently and for longer durations in the
coming decades [74], which based on the results of this work, may negatively impact the early
life stages of *M. pyrifera*. What remains unknown is whether germlings that survive in these
elevated conditions will continue normal development when temperatures return to normal
or if in fact prior exposure may aid in promoting thermal resilience.

## Conclusion

Although this study was conducted on only one population in southern New Zealand, the find-
ings highlight the importance of temperature stress as a controlling factor in the persistence of
*M. pyrifera* populations. Key findings include that; elevated temperatures stimulated greater
spore release during the process of sporulation, spore settlement, germination, and germ-tube
length were negatively impacted by increased temperature and significantly declined above

23.8˚C, 21.7˚C, and 19.8˚C, respectively. This is also the first report to quantify the thermal threshold of spore and germling life stages of *M. pyrifera*. This information is key for making predictions of how such a valuable ecosystem engineer will perform in decades to come. It also offers information that may help inform restoration and aquaculture efforts for this species. Future work should focus on understanding the longevity and viability of haploid stages that have undergone thermal stress, specifically whether such expose has positive or negative effects on the fitness of later developmental stages under normal and elevated temperature regimes. This information is needed to provide insight into the recovery potential of kelp forests post heatwave events.

## Acknowledgments

We would like to thank Namrata Chan, Ben Williams, and Niall Pearson for collecting sporophylls of *M. pyrifera* for this study. We also thank the two anonymous reviewers for their careful and detailed comments and suggestions to improve greatly this manuscript.

## Author Contributions

**Conceptualization:** Mathew J. Desmond, Christopher D. Hepburn.

**Data curation:** Duong M. Le.

**Formal analysis:** Duong M. Le, Daniel W. Pritchard.

**Funding acquisition:** Mathew J. Desmond, Christopher D. Hepburn.

**Investigation:** Duong M. Le.

**Methodology:** Duong M. Le, Mathew J. Desmond.

**Project administration:** Duong M. Le, Mathew J. Desmond.

**Software:** Duong M. Le, Daniel W. Pritchard.

**Supervision:** Mathew J. Desmond, Daniel W. Pritchard, Christopher D. Hepburn.

**Validation:** Mathew J. Desmond, Christopher D. Hepburn.

**Visualization:** Duong M. Le, Daniel W. Pritchard.

**Writing – original draft:** Duong M. Le, Mathew J. Desmond.

**Writing – review & editing:** Duong M. Le, Mathew J. Desmond, Daniel W. Pritchard, Christopher D. Hepburn.

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
