## [Decision Letter · Decision Letter 0]

15 Jul 2022

PONE-D-22-07775Effect of temperature on sporulation and spore development of giant kelp (*Macrocystis pyrifera*)PLOS ONE

Dear Dr. Le,

Thank you for submitting your manuscript to PLOS ONE. After careful consideration, we feel that it has merit but does not fully meet PLOS ONE’s publication criteria as it currently stands. Therefore, we invite you to submit a revised version of the manuscript that addresses the points raised during the review process.

We look forward to receiving your revised manuscript.

Kind regards,

Rui Rosa

Academic Editor

PLOS ONE

Journal Requirements:

Reviewers' comments:

Reviewer's Responses to Questions

**Comments to the Author**

1. Is the manuscript technically sound, and do the data support the conclusions?

Reviewer #1: Yes

Reviewer #2: Yes

2. Has the statistical analysis been performed appropriately and rigorously? 

Reviewer #1: Yes

Reviewer #2: Yes

3. Have the authors made all data underlying the findings in their manuscript fully available?

Reviewer #1: Yes

Reviewer #2: No

4. Is the manuscript presented in an intelligible fashion and written in standard English?

Reviewer #1: Yes

Reviewer #2: Yes

5. Review Comments to the Author

Reviewer #1: This manuscript examines the effects of temperature on giant kelp (M. pyrifera) in its earliest, microscopic life stages. The authors found a significant increase in spore release with increasing temperatures, but they also found the growth rate of germlings significantly decreased with increasing temperatures. Overall, this manuscript offers insight into which life stages may be vulnerable to rising ocean temperatures and what mechanisms may drive kelp loss in a changing ocean climate. This work will be particularly helpful for future conservation and restoration efforts for managers hoping to maintain kelp populations in heat-stressed regions.

My primary comments for the manuscript are the following:

- During revision, this manuscript should undergo a grammatical review. There are multiple instances of missing or misused punctuation. Other sentences are missing conjunctions (e.g., “and”) or articles (e.g., “the”). It does not detract from the overall clarity of the manuscript, but it is somewhat distracting.

- In the discussion section, the authors should add language to consider the potential effects of elevated temperature on subsequent life stages. If they are using their findings to advocate for the selection of thermally-tolerant individuals, there should be additional studies cited that support this approach, particularly in aquaculture settings, and they should aim to demonstrate that the effects of thermal stress will not be carried forward as the individual continues to grow.

- In the conclusion (or the discussion section), the authors should also spend more time placing their findings in a broader context of global kelp populations and the stress they face from increasing ocean temperatures.

Please see my additional line-by-line comments below:

Line 28 - The phrase in the abstract should read “spore settlement was the developmental phase most drastically effected by increasing temperature.”

Line 49-52 – The sentence beginning with “Temperature plays a pivotal role…” is a run-on sentence. I would suggest revising it to be two separate sentences.

Line 87 – Could you provide a bit more description about the site from which kelp was sampled? Depth/oceanographic seasons/temperature ranges/etc. Is this a region with abundant, constant, or flashy kelp populations?

Line 103 – Can you please add in a brief justification for the temperature range used in this experiment? Is this the current/projected temperature range for M. pyrifera populations in NZ? (Could re-use citations from Lines 64/65.)

Line 165 – I would also recommend adding in an explicit mention of the model structures. Did you begin with an interaction term for the fixed effects in every model? The interaction is often mentioned in the results, and appears in Table 1, so the interaction effect should first be mentioned in this paragraph if that was the default structure.

Line 168 – Where will this dataset be made available?

Figures 1 & 2 – The difference in spore production and settlement between the warmest treatment and the rest is quite striking!

For Figures 1-4, you might consider making it a single, four-paneled figure, so that readers could compare the outcomes at various stages in the life cycle across temperature treatments. I think they could certainly still be kept separate, but just a thought.

Line 213 – After a mention of the greatest increase in diameter, you should also mention the greater tube diameters measured on Day 1 for temperatures between 12 and 19C. These all appear noticeably larger than the 9 and 11C treatments, not to mention the warmest treatments (23-26C).

Table 1 – Why did you choose to keep Day and Temperature*Day effects in the Germination model when they do not appear to be significant? It seems the other models were trimmed to include only significant fixed effects, so if you have a study design or scientific basis for keeping non-significant effects in, I would add those justifications to the Germination results section.

Line 242 – Isn’t increased spore production also potentially a compensatory response of kelp to counteract the likelihood of increased mortality as a result of increased temperatures? Could you discuss/present examples of this kind of response in other algae/marine species?

Line 265 – If you propose using this strategy for selection of thermally tolerant individuals, you should also discuss the possibility of heat damaging cells prior to dropping temperature back to “normal” levels. If they are damaged during the initial, warmer period, additional research is necessary to determine if they are still viable for germination.

Line 271 – Similar to my comment above, if you advocate for choosing thermally-tolerant individuals, you must also better explain/justify how that tolerance may carry over into other life stages.

Line 287 – The information in the sentence beginning with “Temperature above 19.8C…” should be presented much earlier in the discussion so the reader can be reminded of the biologically relevant temperature range as they read through the remainder of the discussion.

Line 291 – I would like to see some more discussion of the conditions in southern New Zealand from where these kelp were sampled (see my comment for the Methods/Site section), and whether this area is subject to a wide range in temperatures or rapidly rising temperatures, buffered by upwelling, etc. More language describing how this population might relate to other global kelp populations would help to better place this study in a broader context.

Reviewer #2: Effect of temperature on sporulation and spore development of giant kelp (*Macrocystis pyrifera*)

Duong M. Le et al.

The research reports the result of studies aimed at quantifying the effects of temperature on the early developmental stages (survivorship, germination, and growth) of *M. pyrifera*.

I enjoyed reading the well-executed research. The experiments are targeted and precise, and give a good view into the thermal tolerance properties of the various developmental stages (incl. of sporulation). It was nice to again see a good quality old-fashioned studied that thoroughly addressed some fundamental biological properties of kelp.

Some specific comments below:

**Specific**

L.41: I don’t think the “However, ” belongs there and it can be safely removed.

L.64-65: replace “between” with “from” so as to imply that the temperature ranges are inclusive of the temperatures mentioned (and elsewhere in the text).

L.65: “The ambiguity surrounding the of maximum thermal threshold … ” I am not sure there is an ambiguity. Klaus Lüning mentions different kinds of limits, including lethal limits, reproductive limits, and growth limits. The different thermal ranges probably reflect the growth and lethal limits, respectively. But you are right, local adaptation and genetic influences will be part of the explanation.

L.72: “These early life stages may hold important information that will provide a greater understanding of how rising temperature is negatively influencing *M. pyrifera* and an insight into how populations will fare under predicted climate change scenarios.” – definitely. This talks to Lüning’s reproductive limits.

L.88-91: What was the temperature at time of collection? Depth and nutrient concentrations at the time will also be valuable information to add.

L.98-101: Intuitively I presume that the linear temperature gradient due to the design of the block prevented randomising the thermal treatments across space. Will other environmental influences such as unintended light gradients etc. have confounded the experimental setup, specifically with regards to the expectation that all other factors beside temperature must be constant over the experimental space?

L.103-105: How stable were the temperatures during the post-stabilisation phase (i.e. during the experiments)?

L.112-113: Leaving the pieces of sori at 12°C for 30 minutes prior to placing them in the vials seems arbitrary. It could equally well have been 16°C or 22°C, or they may have been placed directly from 4°C without an intermediate ‘room temperature’ adjustment into the vials. What justified the choice of 12°C, and how could other temperatures have affected the subsequent experimental results?

L115: Were the vials stirrer/agitated to ensure the zoospores became resuspended in case they settled?

L.126-127: Is this a continuation of the experiment mentioned under “Effect of temperature on spore release,“ or did you set up a new experiment from scratch?

L.129: How did you treat the data collected at days one, three and five? Reported as separate measurements along a time series? I suppose I can wait until I read the results, but I want to know now.

L.136: How did you define “visibly attached”?

L.137-140: I am not sure it is wise to calculate the percentage of spores settled as you did. The two kinds of counts are from very different approaches (physical counts in aliquots of media in a haemocytometer) vs. counts of attached spores seen in photographs of the settlement surface. I am not sure what the solution is, really. I think that the haemocytometer data accurately reflect the amount of spores produced per known surface area of kelp. But settlement/attachment count depends on the proportion of the available settle surface assessed, and you by no means expressed your data as a proportion of the total available space. Sorry, just some thoughts and I have not spent an adequate amount of time thinking trough it…

L.142: Again, how were the data collected at days one, three and five treated?

L.158-168: Nice!

L.165: “and individual was treated as a random effect“ … each of the wells with the vials in the temperature gradient block?

L.194: what is “exposure time”?

6. PLOS authors have the option to publish the peer review history of their article (what does this mean?). If published, this will include your full peer review and any attached files.

Reviewer #1: No

Reviewer #2: No

---

## [Author Response · Author response to Decision Letter 0]

8 Sep 2022

Reviewer #1: 

This manuscript examines the effects of temperature on giant kelp (M. pyrifera) in its earliest, microscopic life stages. The authors found a significant increase in spore release with increasing temperatures, but they also found the growth rate of germlings significantly decreased with increasing temperatures. Overall, this manuscript offers insight into which life stages may be vulnerable to rising ocean temperatures and what mechanisms may drive kelp loss in a changing ocean climate. This work will be particularly helpful for future conservation and restoration efforts for managers hoping to maintain kelp populations in heat-stressed regions.

Thank you, we appreciate your support of this work and the time you have taken to provide these comments, which have improved the manuscript significantly. 

My primary comments for the manuscript are the following:

- During revision, this manuscript should undergo a grammatical review. There are multiple instances of missing or misused punctuation. Other sentences are missing conjunctions (e.g., “and”) or articles (e.g., “the”). It does not detract from the overall clarity of the manuscript, but it is somewhat distracting.

The final revision of this manuscript has been extensively re-reviewed by all authors with a specific eye to grammar. We have taken care to ensure the manuscript is as clear, and grammatically correct, as possible. 

- In the discussion section, the authors should add language to consider the potential effects of elevated temperature on subsequent life stages. If they are using their findings to advocate for the selection of thermally-tolerant individuals, there should be additional studies cited that support this approach, particularly in aquaculture settings, and they should aim to demonstrate that the effects of thermal stress will not be carried forward as the individual continues to grow.

This overall comment seems to be linked to specific points made below (regarding lines 265 and 271). The lifecycle of this kelp species (i.e., an alternating microscopic gametophyte / very large macroscopic sporophyte) make tackling this problem in controlled experimental conditions, somewhat difficult and beyond the scope of this manuscript. However, we appreciate that further elaboration is helpful. Naturally, our focus has been on the seaweed literature, which we consider is most relevant to the discussion here. We have highlighted a relevant example (Ladah and Zertuche-González, 2007) and we have added to the Discussion (see responses to specific comments below). To the best of our knowledge, this is the only example where recovery (90% recovery after 8 weeks) of relevant life stages of M. pyrifera to comparable thermal shocks has been reported. 

Beyond this example, we are not aware of other work to understand the ongoing deleterious effects of thermal shock and/or the inheritability of these effects (or, for that matter, the inheritability of any potential thermal tolerance). Overall, we consider the experiments presented here are the “first step” to selecting thermally tolerant strains and more work – including examination of deleterious effects – is needed. Ongoing work in our group is examining exactly these questions, but in the meantime, we have adjusted the Discussion of this manuscript and consider this an improvement. 

- In the conclusion (or the discussion section), the authors should also spend more time placing their findings in a broader context of global kelp populations and the stress they face from increasing ocean temperatures.

We agree and have added a paragraph to the end of the Discussion. We think this is linked to your more specific comment (regarding Line 291) and consider that the addition of this paragraph is both justified (given your comments here and below) and that it improves the manuscript overall.

Please see my additional line-by-line comments below:

Line 28 - The phrase in the abstract should read “spore settlement was the developmental phase most drastically effected by increasing temperature.”

Revised page 2 line 26.

Line 49-52 – The sentence beginning with “Temperature plays a pivotal role…” is a run-on sentence. I would suggest revising it to be two separate sentences. 

Revised page 3 line 49-52.

Line 87 – Could you provide a bit more description about the site from which kelp was sampled? Depth/oceanographic seasons/temperature ranges/etc. Is this a region with abundant, constant, or flashy kelp populations? 

Added page 5 line 100-102.

Line 103 – Can you please add in a brief justification for the temperature range used in this experiment? Is this the current/projected temperature range for M. pyrifera populations in NZ? (Could re-use citations from Lines 64/65.).

Added page 5-6 line 116-119.

Line 165 – I would also recommend adding in an explicit mention of the model structures. Did you begin with an interaction term for the fixed effects in every model? The interaction is often mentioned in the results, and appears in Table 1, so the interaction effect should first be mentioned in this paragraph if that was the default structure. 

All possible combinations of main effects were included (and kept) in the model. Three response variables (Spore production, Spore settlement and Survivorship) have only a single predictor (Temperature) because these variables were not measured over time. The remaining response variables (Germination and Germ-tube growth) were measured over Time, and so both this and an interaction with Temperature were included. 

In reviewing your comment here, we now realise that this information, while present, was not as clear as it could have been, and we have revised the Methods (see section Statistical analyses, page 8 line 186) accordingly. 

Line 168 – Where will this dataset be made available? 

Following PLOS One submission guidelines, we have deposited our data in figshare (https://doi.org/10.6084/m9.figshare.20960476.v1). We understand this link will be included with the manuscript during typesetting. 

Figures 1 & 2 – The difference in spore production and settlement between the warmest treatment and the rest is quite striking! 

We agree and consider this is one of the important findings of this research!

For Figures 1-4, you might consider making it a single, four-paneled figure, so that readers could compare the outcomes at various stages in the life cycle across temperature treatments. I think they could certainly still be kept separate, but just a thought

We agree and have switched to a 4-paneled figure. Page 9 line 211.

Line 213 – After a mention of the greatest increase in diameter, you should also mention the greater tube diameters measured on Day 1 for temperatures between 12 and 19C. These all appear noticeably larger than the 9 and 11C treatments, not to mention the warmest treatments (23-26C). 

Added page 10 line 255.

Table 1 – Why did you choose to keep Day and Temperature*Day effects in the Germination model when they do not appear to be significant? It seems the other models were trimmed to include only significant fixed effects, so if you have a study design or scientific basis for keeping non-significant effects in, I would add those justifications to the Germination results section. 

See comments above (Regarding Line 165 in the original manuscript). We consider that our revisions to the Methods section should make it clear that all possible main effects and interactions were kept.

Line 242 – Isn’t increased spore production also potentially a compensatory response of kelp to counteract the likelihood of increased mortality as a result of increased temperatures? Could you discuss/present examples of this kind of response in other algae/marine species?

There are very few published examples of temperature control of spore release in macroalgae generally and to the best of our knowledge none the do the work required to prove a compensatory response. Our work provides some insight, but there remain several unknowns. We have revised our Discussion to acknowledge this lack of knowledge (page 12, line 322) and now cite a large review by Bettignies et al 2018, which reports only 3 relevant examples, from a pool of 81 manuscripts. 

Despite this, your comment has encouraged us to expand our speculation on possible population-level effects for M. pyrifera of elevated spore production (page 12, lines 320). We very deliberately say there are “at least” two potential scenarios (page 12, line 323) and present two that are supported by our data (page 12, lines 323-328). We can image at least one other scenario – namely that elevated spore release might compromise future spore production in more favourable conditions (to the net detriment to M. pyrifera populations) – but clearly more work is required to verify this. 

Line 265 – If you propose using this strategy for selection of thermally tolerant individuals, you should also discuss the possibility of heat damaging cells prior to dropping temperature back to “normal” levels. If they are damaged during the initial, warmer period, additional research is necessary to determine if they are still viable for germination. 

We do agree that further research is required, and we are undertaking some of this research within our group, currently. To address this specific comment we have further highlighted the work of (Ladah and Zertuche-González, 2007), where recovery (up to 90% after 8 weeks) of the haploid life stage of M pyrifera was measured (see page 13, line 342). 

Line 271 – Similar to my comment above, if you advocate for choosing thermally-tolerant individuals, you must also better explain/justify how that tolerance may carry over into other life stages. 

We understand and appreciate this suggestion. In short: It is difficult, because there is a lack of relevant literature in general and we are not aware of any literature at all for M pyrifera. We do now refer directly to this lack of knowledge in the newly-inserted final paragraph (see page 14, line 367 –378) and our group is working to address these questions directly for this species. 

Line 287 – The information in the sentence beginning with “Temperature above 19.8C…” should be presented much earlier in the discussion so the reader can be reminded of the biologically relevant temperature range as they read through the remainder of the discussion.

We agree and have moved this to the first paragraph in the Discussion (page 11 line 301-303).

Line 291 – I would like to see some more discussion of the conditions in southern New Zealand from where these kelp were sampled (see my comment for the Methods/Site section), and whether this area is subject to a wide range in temperatures or rapidly rising temperatures, buffered by upwelling, etc. More language describing how this population might relate to other global kelp populations would help to better place this study in a broader context.

We have added a paragraph to the end of the Discussion which includes specific local information and acknowledgement that our experiments apply to other temperate regions with M. pyrifera populations, many of which are experiencing similar changes (page 14 line 367-378).

References:

Ladah, Lydia B., and Zertuche-González, José A., 2007. Survival of microscopic stages of a perennial kelp (Macrocystis pyrifera) from the center and the southern extreme of its range in the Northern Hemisphere after exposure to simulated El Niño stress. Mar Biol (2007) 152:677–686 DOI 10.1007/s00227-007-0723-z 

de Bettignies T, Wernberg T and Gurgel CFD (2018). Exploring the influence of temperature on aspects of the reproductive phenology of temperate seaweeds. Front. Mar. Sci. 5:218. doi: 10.3389/fmars.2018.00218 

 

Reviewer #2: Effect of temperature on sporulation and spore development of giant kelp (Macrocystis pyrifera)

Duong M. Le et al.

The research reports the result of studies aimed at quantifying the effects of temperature on the early developmental stages (survivorship, germination, and growth) of M. pyrifera. I enjoyed reading the well-executed research. The experiments are targeted and precise, and give a good view into the thermal tolerance properties of the various developmental stages (incl. of sporulation). It was nice to again see a good quality old-fashioned studied that thoroughly addressed some fundamental biological properties of kelp.

We appreciate the support of our work. We agree that these “old fashioned” studies remain a powerful tool for exploring these fundamental questions. We appreciate the time taken to review this manuscript and have addressed your specific comments below. 

Some specific comments below:

L.41: I don’t think the “However, ” belongs there and it can be safely removed. 

Removed page 3 line 41.

L.64-65: replace “between” with “from” so as to imply that the temperature ranges are inclusive of the temperatures mentioned (and elsewhere in the text). 

Replaced page 4 line 70, 71; page 9 line 224.

L.65: “The ambiguity surrounding the of maximum thermal threshold … ” I am not sure there is an ambiguity. Klaus Lüning mentions different kinds of limits, including lethal limits, reproductive limits, and growth limits. The different thermal ranges probably reflect the growth and lethal limits, respectively. But you are right, local adaptation and genetic influences will be part of the explanation. 

We have removed the term “ambiguity” (page 4, line 71). 

L.72: “These early life stages may hold important information that will provide a greater understanding of how rising temperature is negatively influencing M. pyrifera and an insight into how populations will fare under predicted climate change scenarios.” – definitely. This talks to Lüning’s reproductive limits. 

We agree and appreciate your support of this work. 

L.88-91: What was the temperature at time of collection? Depth and nutrient concentrations at the time will also be valuable information to add.

We have added temperature and depth information (page 5, line 100-102). Nutrient data form the site at the time of collection are not available but given the collection time (October / Austral Spring) we expect they will be likely non-limiting. 

L.98-101: Intuitively I presume that the linear temperature gradient due to the design of the block prevented randomising the thermal treatments across space. Will other environmental influences such as unintended light gradients etc. have confounded the experimental setup, specifically with regards to the expectation that all other factors beside temperature must be constant over the experimental space?

Yes, you are right, the linear temperature was due to block design. Light was at 20.3 (±1.7) µmol photons m-2 s-1 depending on the location of vials. We have added this information in revision page 6 line 121-122.

L.103-105: How stable were the temperatures during the post-stabilisation phase (i.e. during the experiments)?

The temperature varied about 0.2°C. This information is included on page 5, line 116.

L.112-113: Leaving the pieces of sori at 12°C for 30 minutes prior to placing them in the vials seems arbitrary. It could equally well have been 16°C or 22°C, or they may have been placed directly from 4°C without an intermediate ‘room temperature’ adjustment into the vials. What justified the choice of 12°C, and how could other temperatures have affected the subsequent experimental results? 

This was done to bring tissue to a temperature more similar to the treatment conditions before immersing them (rather than directly from the refrigerator at 4°C). 12°C was simply the ambient temperature of the room and by stating this we acknowledge all conditions the tissue was exposed to. The effect of initial temperature remains unknown but was consistent for all treatments. 

L115: Were the vials stirrer/agitated to ensure the zoospores became resuspended in case they settled?

Yes, this has now been clarified Page 6 line 131.

L.126-127: Is this a continuation of the experiment mentioned under “Effect of temperature on spore release,” or did you set up a new experiment from scratch?

This was a continuation. This is now clearer with revisions to this section, now on page 7 line 148 – 150. 

L.129: How did you treat the data collected at days one, three and five? Reported as separate measurements along a time series? I suppose I can wait until I read the results, but I want to know now.

All treatments were analysed as factors (categories). This is addressed in a dedicated section (Statistical analyses, page 8 line 186) because there are some commonalities across the analysis of the response variables. 

L.136: How did you define “visibly attached”?

We have removed the word “visibly” page 7 line 160. This word is no longer needed as we now provide a more precise description of our methods (see page 7, lines 162 to 163)

L.137-140: I am not sure it is wise to calculate the percentage of spores settled as you did. The two kinds of counts are from very different approaches (physical counts in aliquots of media in a haemocytometer) vs. counts of attached spores seen in photographs of the settlement surface. I am not sure what the solution is, really. I think that the haemocytometer data accurately reflect the amount of spores produced per known surface area of kelp. But settlement/attachment count depends on the proportion of the available settle surface assessed, and you by no means expressed your data as a proportion of the total available space. Sorry, just some thoughts and I have not spent an adequate amount of time thinking trough it

We are confident that we have calculated this correctly. In considering your comment, we have realised that some small revisions could improve the clarity of the methods. We are now clear that our calculations are predicated on an assumption that all spore that could have settled potentially settled would have done so 24 hours after release (page 7, line 148-150) and introduce the phrase “maximum potential settlement” (page 7, line 162) to better link this to the description of percentage settlement (page 7, line 160-163). 

As you note, another implicit assumption is that all spores that could settle would have space to do so. This is not stated explicitly in the manuscript, but we consider it a reasonable assumption given the relatively low proportion of basal area (maximum 64%) of the vial that was covered in settled cells. 

We consider that this is an improvement and appreciate you drawing our attention to this. 

L.142: Again, how were the data collected at days one, three and five treated? 

See response to comment above (Regarding line L.129 of the original manuscript). 

L.158-168: Nice!

See response to comment above (Regarding line L.129 of the original manuscript). As described previously, we consider this is the best place for this information. 

L.165: “and individual was treated as a random effect“ … each of the wells with the vials in the temperature gradient block?

No. Sori from six independent M. pyrifera individuals were used in this study. This is detailed in the first section of the Methods (Site selection and tissue collection) and we are not sure how to revise the wording you have highlighted here to make this more clear. 

L.194: what is “exposure time”? 

We have revised the manuscript so that we consistently use “time” when referring to the predictor we used to refer to experimental days 1, 3 and 5.

---

## [Decision Letter · Decision Letter 1]

14 Nov 2022

Effect of temperature on sporulation and spore development of giant kelp (*Macrocystis pyrifera*)

PONE-D-22-07775R1

Dear Dr. Le,

We’re pleased to inform you that your manuscript has been judged scientifically suitable for publication and will be formally accepted for publication once it meets all outstanding technical requirements.

Kind regards,

Rui Rosa

Academic Editor

PLOS ONE

Additional Editor Comments (optional):

Reviewers' comments:

Reviewer's Responses to Questions

**Comments to the Author**

1. If the authors have adequately addressed your comments raised in a previous round of review and you feel that this manuscript is now acceptable for publication, you may indicate that here to bypass the “Comments to the Author” section, enter your conflict of interest statement in the “Confidential to Editor” section, and submit your "Accept" recommendation.

Reviewer #1: All comments have been addressed

2. Is the manuscript technically sound, and do the data support the conclusions?

Reviewer #1: Yes

3. Has the statistical analysis been performed appropriately and rigorously? 

Reviewer #1: Yes

4. Have the authors made all data underlying the findings in their manuscript fully available?

Reviewer #1: Yes

5. Is the manuscript presented in an intelligible fashion and written in standard English?

Reviewer #1: Yes

6. Review Comments to the Author

Reviewer #1: The authors have significantly revised their previous submission, and I believe the text is much better for it. They have done a good job of incorporating both sets of comments, and I have included only minor edits to the existing text below. I look forward to hearing about additional such studies from this group in the coming years!

Line 57 – The phrase should read “mesograzers, bryozoans, and sea urchins.”

Line 58 – Be sure to place a comma after the “[20]” citation to separate the introductory phrase (“by the turn of the century [20],”).

Line 61 – The beginning of the sentence should read “Macrocystis pyrifera is arguably one of the most iconic kelps, creating the largest biogenic system of all …”

Line 76-77 – The phrase should read “how rising temperatures are negatively influencing M. pyrifera and insight into how…”.

Line 104 – To help address the other reviewer’s comment regarding the individual as random effect model structure, perhaps you could add the following phrase to the end of this sentence: “to allow for six replicates at each temperature level, one for each of the individual sori collected.”

Line 111 – Good justification for thermal range selection.

Line 181 – Prior to determination of significant effects, how did you diagnose model fit (i.e., inspection of residuals, etc.)?

Figure 1 – This looks much better as a 4 paneled figure, and the reader can much more clearly compare trends across life stages.

Line 273 – I would advocate for removing the parentheses from this sentence and splitting it in two. So, it would read as follows: “At least two potential scenarios exist, the first being that elevated temperatures cause the mass release of spores which are negatively impacted by elevated temperatures, with no net benefit to M. pyrifera populations. A second scenario may be that increased temperature transiently increases spore delivery to the substrate under a scenario where temperature then drops back below harmful thresholds, which might be a net benefit to M. pyrifera recruitment.”

Line 312 – The phrases “germ-tube,” “germ tube,” and “germination tubes” are used interchangeably. I would advise choosing a uniform phrasing and reviewing your use of it throughout the manuscript.

Line 318 – This paragraph provides very helpful context for the study.

Line 326 – In addition to “normal development” and “thermal resilience” scenarios, you might also acknowledge the potential of germlings to fare worse if they survive increased temperatures during early developmental stages.

Line 331 – Somewhere in the final conclusion paragraph, you should briefly restate the study’s primary findings regarding sporulation, settlement, germination, and germ-tube development.

Line 334 – You might also consider adding to this sentence to reiterate the need for additional study of these earliest life stages.

7. PLOS authors have the option to publish the peer review history of their article (what does this mean?). If published, this will include your full peer review and any attached files.

Reviewer #1: No

---

## [Editor Report · Acceptance letter]

18 Nov 2022

PONE-D-22-07775R1 

Effect of temperature on sporulation and spore development of giant kelp (*Macrocystis pyrifera*) 

Dear Dr. Le:

I'm pleased to inform you that your manuscript has been deemed suitable for publication in PLOS ONE. Congratulations! Your manuscript is now with our production department. 

Kind regards, 

on behalf of

Dr. Rui Rosa 

Academic Editor

PLOS ONE